# Proof-of-Concept for the Analgesic Effect and Thermoregulatory Safety of Orally Administered Multi-Target Compound SZV 1287 in Mice: A Novel Drug Candidate for Neuropathic Pain

**DOI:** 10.3390/biomedicines9070749

**Published:** 2021-06-29

**Authors:** Ádám István Horváth, Nikolett Szentes, Valéria Tékus, Maja Payrits, Éva Szőke, Emőke Oláh, András Garami, Eszter Fliszár-Nyúl, Miklós Poór, Cecília Sár, Tamás Kálai, Szilárd Pál, Krisztina Percze, Éva Nagyné Scholz, Tamás Mészáros, Blanka Tóth, Péter Mátyus, Zsuzsanna Helyes

**Affiliations:** 1Department of Pharmacology and Pharmacotherapy, Medical School, University of Pécs, H-7624 Pécs, Hungary; adam.horvath@aok.pte.hu (Á.I.H.); szentes.nikolett@gmail.com (N.S.); valeria.tekus@aok.pte.hu (V.T.); payrits.maja@gmail.com (M.P.); eva.szoke@aok.pte.hu (É.S.); 2Molecular Pharmacology Research Group & Centre for Neuroscience, János Szentágothai Research Centre, University of Pécs, H-7624 Pécs, Hungary; 3ALGONIST Biotechnologies GmBH, 1030 Vienna, Austria; 4Department of Thermophysiology, Institute for Translational Medicine, Medical School, University of Pécs, H-7624 Pécs, Hungary; emoke.olah@aok.pte.hu (E.O.); andras.garami@aok.pte.hu (A.G.); 5Department of Pharmacology, Faculty of Pharmacy, University of Pécs, H-7624 Pécs, Hungary; eszter.nyul@aok.pte.hu (E.F.-N.); poor.miklos@pte.hu (M.P.); 6Institute of Organic and Medicinal Chemistry, Faculty of Pharmacy, University of Pécs, H-7624 Pécs, Hungary; cecilia.sar@aok.pte.hu (C.S.); tamas.kalai@aok.pte.hu (T.K.); 7Institute of Pharmaceutical Technology and Biopharmacy, Faculty of Pharmacy, University of Pécs, H-7624 Pécs, Hungary; pal.szilard@gytk.pte.hu; 8Department of Molecular Biology, Institute of Biochemistry and Molecular Biology, Faculty of Medicine, Semmelweis University, H-1094 Budapest, Hungary; k.percze@gmail.com (K.P.); scholz1eva@yahoo.co.uk (É.N.S.); meszaros.tamas@med.semmelweis-univ.hu (T.M.); 9Department of Inorganic and Analytical Chemistry, Budapest University of Technology and Economics, H-1111 Budapest, Hungary; tblanka@mail.bme.hu; 10Institute of Digital Health Sciences, Faculty of Health and Public Services, Semmelweis University, H-1094 Budapest, Hungary; peter.maty@gmail.com; 11PharmInVivo Ltd., H-7629 Pécs, Hungary

**Keywords:** semicarbazide-sensitive amine oxidase, vascular adhesion protein-1, partial sciatic nerve ligation, traumatic neuropathy, SZV 1287, transient receptor potential vanilloid 1, drug development

## Abstract

SZV 1287 (3-(4,5-diphenyl-1,3-oxazol-2-yl)propanal oxime) is a novel multi-target candidate under preclinical development for neuropathic pain. It inhibits amine oxidase copper containing 3, transient receptor potential ankyrin 1 and vanilloid 1 (TRPV1) receptors. Mainly under acidic conditions, it is transformed to the cyclooxygenase inhibitor oxaprozin, which is ineffective for neuropathy. Therefore, an enterosolvent capsule is suggested for oral formulation, which we investigated for nociception, basic kinetics, and thermoregulatory safety in mice. The antihyperalgesic effect of SZV 1287 (10, 20, 50, and 200 mg/kg, p.o.) was determined in partial sciatic nerve ligation-induced traumatic neuropathy by aesthesiometry, brain and plasma concentrations by HPLC, and deep body temperature by thermometry. Its effect on proton-induced TRPV1 activation involved in thermoregulation was assessed by microfluorimetry in cultured trigeminal neurons. The three higher SZV 1287 doses significantly, but not dose-dependently, reduced neuropathic hyperalgesia by 50% of its maximal effect. It was quickly absorbed; plasma concentration was stable for 2 h, and it entered into the brain. Although SZV 1287 significantly decreased the proton-induced TRPV1-mediated calcium-influx potentially leading to hyperthermia, it did not alter deep body temperature. Oral SZV 1287 inhibited neuropathic hyperalgesia and, despite TRPV1 antagonistic action and brain penetration, it did not influence thermoregulation, which makes it a promising analgesic candidate.

## 1. Introduction

Neuropathic pain of 3–17% prevalence represents a great unmet medical need, since the conventional and adjuvant analgesics are frequently ineffective, and they exert the broad range of severe adverse effects upon long-term use [1,2,3,4]. Therefore, it is a pressing need to clarify its pathophysiology and develop analgesics with novel modes of action.

Distinctly, our novel, multi-target drug candidate SZV 1287 (3-(4,5-diphenyl-1,3-oxazol-2-yl) propanal oxime) offers a substantial breakthrough for the treatment of neuropathic pain [5]. SZV 1287 is an oxime analogue of the cyclooxygenase (COX) inhibitor oxaprozin, which has been used for a long time as a classical non-steroidal anti-inflammatory drug with the indication of osteoarthritis (OA) and rheumatoid arthritis (RA) [6,7]. SZV 1287 itself is an irreversible amine oxidase copper containing 3 (AOC3) inhibitor, and it is metabolized mainly to the COX inhibitor oxaprozin throughout the body [8]. Therefore, SZV 1287 is considered to be the “active prodrug” of oxaprozin [8,9]. Furthermore, we have recently shown that, independently of its AOC3 inhibitory action, SZV 1287 directly antagonizes the pain-sensing transient receptor potential vanilloid 1 (TRPV1) and ankyrin 1 (TRPA1) receptors, which are non-selective cation channels on nociceptive primary sensory neurons [10].

AOC3 is a multifunctional molecule, which belongs to the semicarbazide-sensitive amine oxidase (SSAO) enzyme family. Its membrane-bound form was discovered in 1998, when it was described to be the same as the vascular adhesion protein-1 (VAP-1) after identifying the amino acid sequence and the enzymatic activity of the protein [11]. AOC3 is primarily expressed on endothelial, vascular smooth muscle and adipose cells, but it can be also found in soluble form in the plasma [12]. It produces potentially irritating aldehydes, such as formaldehyde, methylglyoxal, and acrolein as well as hydrogen peroxide and ammonia from primary amines, such as methylamine, aminoacetone, and spermine [13,14]. These products mediate inflammatory and pain mechanisms including TRPA1 activation on sensory nerves, immune, and endothelial cells [15,16]. Furthermore, AOC3 also acts as an adhesive molecule mediating leukocyte transmigration from the postcapillary venules into inflamed areas and promote neovascularization [14]. Thus, it is not a surprise that AOC3 plays an important role in several inflammatory and inflammation-associated diseases, such as RA, bronchial asthma, chronic obstructive pulmonary disease, Alzheimer’s disease, multiple sclerosis, diabetic angiopathy, atherosclerosis, as well as inflammatory liver and eye diseases [17]. The anti-inflammatory and anti-angiogenic actions of small-molecule AOC3 inhibitors and anti-AOC3 antibodies were demonstrated in models of these diseases [14]. Moreover, several AOC3 inhibitors (ASP-8232, PXS-4728A, PRX-167700, and TERN-201) and one antibody (BTT-1023/timolumab) are currently investigated in clinical studies with indications of diabetic macular edema, diabetic nephropathy, diabetic retinopathy, non-alcoholic steatohepatitis, OA, and primary sclerosing cholangitis [18].

We were the first to describe the role of AOC3 in pain processing and sensitization, as well as the antihyperalgesic effects of AOC3 inhibitors in different pain models [19,20]. These effects are at least partially mediated by decreased production of highly reactive AOC3 products activating the TRPA1 [20]. Based on this action and direct dual TRPA1 and TRPV1 antagonism, SZV 1287 was selected to be the lead molecule of our novel analgesic development project. The anti-inflammatory effect of SZV 1287 was previously shown in several mechanism models of inflammation [21]; however, we also showed its antihyperalgesic effect following intraperitoneal (i.p.) administration in two different arthritis models and in an inflammation-independent traumatic neuropathy model [19,20], in which its main metabolite, oxaprozin, was ineffective [20]. The ability of SZV 1287 to inhibit neuropathic mechanical hyperalgesia suggests that it exerts a central mechanism of action, which was recently supported by in vivo dynamic positron emission tomography/magnetic resonance (PET/MR) imaging results demonstrating rapid blood–brain barrier penetration [22].

On the basis of proof-of-concept results showing the efficacy of i.p. SZV 1287 for a broad range of nociceptive mechanisms, we initiated the drug development process with the main indication of neuropathic pain. Here, we demonstrated the antihyperalgesic effect and the thermoregulatory safety as well as provided quantitative data for absorption and brain penetration abilities of this analgesic candidate in the oral formulation aimed to be used in the clinical trials.

## 2. Materials and Methods

### 2.1. Synthesis of SZV 1287

SZV 1287 (3-(4,5-diphenyl-1,3-oxazol-2-yl)propanal oxime) and its four main metabolites (3-(4,5-diphenyloxazol-2-yl)propanoic acid (oxaprozin), 3-(4-(4-hydroxyphenyl)-5-phenyloxazol-2-yl)propanoic acid (L 2799), 3-(5-(4-hydroxyphenyl)-4-phenyloxazol-2-yl)propanoic acid (L 2805), and 3-(4,5-bis(4-hydroxyphenyl)oxazol-2-yl)propanoic acid (L 2811)) were first synthesized at the Department of Organic Chemistry, Faculty of Pharmacy, Semmelweis University, Budapest, Hungary. The scaling-up process was then developed at the Institute of Organic and Medicinal Chemistry, Faculty of Pharmacy, University of Pécs, Pécs, Hungary. For this study, it was provided as previously described [9]. Physicochemical characteristics of SZV 1287 and its metabolites including nuclear magnetic resonance (NMR) and mass spectrometry data, as well as the melting point were previously published [23,24].

### 2.2. Preparation of Enterosolvent SZV 1287 Capsules

SZV 1287 mixed with the filler excipient lactose monohydrate (Molar Chemicals, Halásztelek, Hungary) were filled into “M”-size hard gelatin capsules (Torpac, Fairfield, NJ, USA) containing 0.4 mg SZV 1287 corresponding to the 10 mg/kg, 0.8 mg to the 20 mg/kg, and 2 mg to the 50 mg/kg for a mouse weighing 40 g in the nociceptive study. In the case of the 200 mg/kg doses, four capsules containing 2 mg SZV 1287 were administered. In the thermophysiological study, the capsules contained 0.5 mg SZV 1287 corresponding to the 20 mg/kg and 1.25 mg to the 50 mg/kg for a mouse weighing 25 g. All ingredients were accurately weighed on analytical balance (Kern ABJ 220-4M, Kern & Sohn GmbH, Balingen, Germany) after pre-treatment sieving through a 40-mesh sieve. Stirring of powder blends with pestle in mortar was carried out for 10 min, ultimately mixing the completed blend. The flow properties of final blends were considered to be suitable for capsule filling, which was carried out by the manufacturer’s capsule filling device (Torpac, Fairfield, NJ, USA).

Enterosolvent (acid resistant) coating of SZV 1287 capsules was required on the basis of the acid sensitivity of SZV 1287 demonstrated by the proton (^1^H) NMR study (Appendix A). Methodological details of NMR spectra recorded in acidic media are described in Section A.1. After closing the capsules, enterosolvent coating was applied on the surface by the immersion technique. During the film coating, aqueous dispersion of the Eudragit L30D-55 (Evonik Industries, Essen, Germany) film-forming polymer was created with triethyl-citrate (Molar Chemicals, Halásztelek, Hungary), purified water, and talc (Molar Chemicals, Halásztelek, Hungary) using an Ultra-turrax high-shear homogenizer (IKA Werke, Staufen im Breisgau, Germany). The dispersion was filled into a square bath equipped with a magnetic stirrer. Filled capsules were inserted into a tight perforated plate until the middle height of the capsules. After inserting all the capsules, the plate was placed into the dispersion bath containing the coating medium, immersing the bottom part of the capsules for 5 s. Then, the plate was turned upside down, and the upper part was also immersed into the coating liquid bath for another 5 s. After immersion of both sides, 120 min of 40 °C circulating warm air drying was applied to form and congeal the film coating around the capsules. The coating procedure was repeated one more time to ensure the proper coating on the whole capsule surface. After complete coating, 24-h 25 °C drying was the final step of the operation. The dynamics of SZV 1287 release of the enterosolvent capsule at an acidic pH of 1.2 followed by a near-neutral pH of 6.8 is demonstrated in Appendix A. The in vitro dissolution test of enterosolvent SZV 1287 capsules is described in Section A.2. Non-coated placebo capsules containing lactose monohydrate were also prepared.

### 2.3. Mouse Experiments

#### 2.3.1. Animals

The nociceptive and thermophysiological studies were performed on 8–12-week-old male NMRI (weighing 35–45 g) and C57BL/6J mice (weighing 20–30 g), respectively. These body weight ranges refer to the whole experimental series; the majority of mice weighed 38–42 g and 23–26 g, respectively, and only 1 or 2 animals per group were less or more. Mice were bred in the Laboratory Animal House of the Department of Pharmacology and Pharmacotherapy, University of Pécs, Medical School. NMRI mice were maintained in the Laboratory Animal House of the János Szentágothai Research Centre, University of Pécs, C57BL/6J mice in the Laboratory Animal House of the Institute for Translational Medicine in 330 × 160 × 137 mm sized cages with a maximum of 5 mice per cage, under a 12-h light/dark cycle at 24 ± 2 °C, with 50–60% humidity. They were provided with standard mouse chow (ssniff Spezialdiäten GmbH, Soest, Germany) and water ad libitum.

#### 2.3.2. Ethics

The study was conducted according to the European legislation (Directive 2010/63/EU) and Hungarian Government regulation (40/2013., II. 14.) regarding the protection of animals used for scientific purposes, in full compliance with to recommendations of the International Association for the Study of Pain. It was approved by the Animal Welfare Committee of the University of Pécs and the National Scientific Ethical Committee on Animal Experimentation of Hungary and licensed by the Government Office of Baranya County (licence No.: BA02/2000–41/2016 and BA02/2000-09/2021).

#### 2.3.3. Nociceptive Study

##### Experimental Design

The optimal time point for the post-treatment mechanonociceptive withdrawal threshold measurement and potential influence of fasting on the peak plasma concentration of SZV 1287 were determined in an exploratory pharmacokinetic study. One group of mice was fasted for 16 h before the oral treatment, the other one was provided with food ad libitum. The food was provided to the fasted group 1 h after the treatment. Each mouse was treated with enterosolvent SZV 1287 capsule (20 mg/kg), then 8 mice per time point were deeply anesthetized with sodium pentobarbital (100 mg/kg, i.p.) 1, 2, 4, 8, and 24 h following the treatment and 150–200 μL blood was taken by cardiac puncture to analyze the plasma concentrations of SZV 1287 and its main metabolites (oxaprozin, L 2799, L 2805, and L 2811).

The dose-dependent effect of a single enterosolvent SZV 1287 capsule treatment was examined on day 7 in the partial sciatic nerve ligation model of traumatic neuropathy. The mechanonociceptive withdrawal threshold of both hind paws was measured before and 7 days after the unilateral nerve ligation prior to drug treatment to determine the difference between the pre- and postoperative thresholds and to prove the development of neuropathic mechanical hyperalgesia in response to the surgery. The effect of SZV 1287 (10, 20, 50, and 200 mg/kg) compared to placebo was determined by repeated mechanonociceptive testing 2 h after the capsule administration. The contralateral intact hind paws were also measured to detect the effect of SZV 1287 on physiological mechanonociception. The oral administration of the capsules was performed by a stainless-steel dosing syringe (X-M Syringe, Torpac, Fairfield, NJ, USA). Since the SZV 1287 content of the specific enterosolvent capsule formulation was calculated for mice weighing 40 g, small dose differences could not be avoided between the individuals belonging to the same group because of the minimal mouse body weight variability within a group. In order to minimize the dosing variability, mice of 40–42 g were only selected for the 200 mg/kg dose. The applied doses of SZV 1287 were selected on the basis of our previous dose-dependency experiment performed with i.p. administration [20]. The animals were not fasted prior to the drug administration since this may have disturbed the behavioral testing, and exploratory measurements showed no significant influence of fasting on the peak plasma concentration of the drug. Following the post-treatment measurements, mice were deeply anesthetized with sodium pentobarbital (100 mg/kg, i.p.), 150–200 μL blood was taken by cardiac puncture, and brains were removed to analyze the plasma and brain concentrations of SZV 1287.

##### Experimental Model

The nociceptive study was carried out in the partial sciatic nerve ligation model of traumatic neuropathy, which is a well-established, highly reproducible, and easily performable model of neuropathic pain characterized by mechanical hyperalgesia [25,26]. It is widely used to examine the effects of novel analgesic candidates for the unmet need of these medical conditions [27,28,29,30].

Partial sciatic nerve ligation was performed in male NMRI mice under ketamine/xylazine (100/5 mg/kg, i.p.) anesthesia, as previously described [25]. Briefly, the ipsilateral common sciatic nerve was exposed high in the thigh region and one-third to one-half of the nerve thickness was tightly ligated using a siliconized silk suture (Ethilon 8-0, Ethicon, Somerville, NJ, USA). The wound was closed with a skin suture (Mersilk 4-0, Ethicon, Somerville, NJ, USA), and the animals were allowed to survive for 7 days, because significant mechanical hyperalgesia develops without postoperative pain component by this time point [20].

##### Measurement of Mechanonociceptive Withdrawal Thresholds of the Hind Paws

Mechanonociceptive withdrawal thresholds of the hind paws were measured by dynamic plantar aesthesiometry (Ugo Basile, Gemonio, Italy), as previously described [20]. Briefly, mice were placed into acrylic glass boxes with metal mesh floors, where they could move around freely. After acclimation, the plantar surface of the hind paw was touched with a blunt-end needle that elevated with a continuously increasing upward force (maximum force of 10 g with 4 s latency) until the animal withdrew its paw. Three consequent measurements per hind paw were performed in each time point, from which mean values were calculated. Mechanonociceptive withdrawal thresholds were expressed in grams (g) and their percentage decrease values compared to the initial pre-surgery controls were presented as mechanical hyperalgesia. Animals with less than 20% pre-treatment hyperalgesia were excluded from the study. Sixtyone mice were originally enrolled in the study to ligate the sciatic nerve, but 14 mice failed to develop mechanical hyperalgesia (a success rate of 77%).

##### Measurement of Plasma and Brain Concentrations of SZV 1287 and its Main Metabolites

The concentrations of SZV 1287 and its four main metabolites (oxaprozin, L 2799, L 2805, and L 2811) were measured by both high-performance liquid chromatography-ultraviolet (HPLC-UV) and HPLC-fluorescence detector (FLD) analyses in the plasma and by HPLC-UV in the brain. The most likely route of formation of SZV 1287 metabolites is demonstrated in Appendix A. In principle, it is also possible that SZV 1287 is hydroxylated first and then the resulting hydroxy derivatives are oxidized to hydroxyoxaprozins (L 2799, L 2805, and L 2811). Blood samples were collected by cardiac puncture into ice-cold tubes containing 8 μL of 50 mg/mL EDTA at a pH of 7.5 (Reanal, Budapest, Hungary) immediately after the post-treatment nociceptive measurement. Then samples were centrifuged for 5 min at 1000 rpm and for 10 min at 10,000 rpm and stored at −80 °C until HPLC analyses. After collecting blood samples, brains were also removed and snap-frozen in liquid nitrogen. The samples were stored at −80 °C until HPLC analyses.

##### HPLC-UV Analyses

Millipore 18 Ohm water was used for all experiments. Potassium-dihidrophosphate (KH_2_PO_4_, 99.6% purity), phosphoric acid cc. (high purity), and acetonitrile (ACN, HPLC gradient grade) were purchased from Molar Chemicals Kft. (Halásztelek, Hungary). All standard substances were stored in powder form at 4 °C.

To prepare the stock solutions, 1 mg of SZV 1287 and all other test substances were separately dissolved in 1 mL gradient-grade ACN. The obtained 1 mg/mL stock solutions were mixed in equal ratio to achieve a 200 µg/mL stock solution that contained all 5 of the test substances. All stock solutions were stored at −20 °C.

Calibration standards were freshly prepared every day by serial dilution. The 10 µL of 200 µg/mL mixed stock solution was added to 190 µL of mouse plasma and serially diluted to achieve a concentration range between 0.5–10 µg/mL for L 2799, 2805, 2811, and 0.5–50 µg/mL for SZV 1287 and oxaprozin. The calibration samples were pre-treated as described below. Prior to the HPLC measurements, proteins of plasma and tissue homogenates were precipitated. In the case of mouse brain tissue, 0.5–1 g of sample was weighed and homogenized in a threefold volume of water by using a UP200St Ultrasonic Homogenizer (Hielscher Ultrasonics GmbH, Teltow, Germany) with C: 50%, Amp: 90% parameters. Fifty µL of tissue homogenate or plasma was pipetted in Eppendorf tubes and completed with 150 µL acetonitrile solution. Following 10 s of vortexing, the samples were centrifuged at 14,000 rpm for 10 min at 4 °C. Forty µL of supernatants were mixed with 60 µL of water in a disposable 200 µL glass vial, and 20 µL of the samples were injected into Kinetex^®^ EVO C18 5 µm 100Å 100 × 3.0 mm (Phenomenex Inc., Torrance, CA, USA) analytical column through SecurityGuard™ ULTRA cartridges for the EVO-C18 (Phenomenex Inc., Torrance, CA, USA) precolumn and were separated by the chromatographic system (Waters Alliance e2695 Separation Module (Waters Corporation, Milford, MA, USA)). The elution was performed with a 0.5 mL/min flow rate at 30 °C using a mixture of ACN and 10 mM KH_2_PO_4_ pH 3.0 (30:70 *v*/*v*%) as eluent A and 100% ACN as eluent B as mobile phases. The following gradient program was applied: 0–1 min: 100% eluent A; 1–6 min: linear change to 86% eluent A, 14% eluent B; 6–12 min: 86% eluent A, 14% eluent B; 12–13 min: linear change to 57% eluent A, 43% eluent B; 15–16 min: linear change to 100% eluent A, 16–21 min: 100% eluent A. The eluted substances were measured at 285 nm by applying a Waters 2489 UV/Visible Detector (Waters Corporation, Milford, MA, USA). The obtained data was analyzed using Empower 3 Chromatography Data Software (Waters Corporation, Milford, MA, USA). Limit of detection (LOD) and limit of quantification (LOQ) values were determined as the lowest concentration where the signal-to-noise ratio was 3 and 10, respectively. The LOD and LOQ values of analyzed substances were 0.2 µg/mL and 0.5 µg/mL, respectively. Six parallel calibrations were measured at 0.5–10 µg/mL (for L 2799, 2805, and 2811) and 0.5–50 µg/mL (for SZV 1287 and oxaprozin) concentration levels to study the linearity of the method. As a matrix, different plasma or brain tissue homogenate batches were used. The calibration curve was fitted on the calibration points using the least squares method. The R^2^ of the curves fitted to plasma calibration samples was 0.999 for SZV 1287, oxaprozin, L 2799, and L 2805 and 0.997 for L 2811. For brain homogenate samples, the R^2^ of calibration curves was 0.998 for all test substances. Within-day repeatability and within-day accuracy values were in the 2.09–6.37% and 100.2–115.2% range, respectively. Recovery of the substances from plasma and brain tissue samples was between 89–116% and 83–112%, respectively, when plasma and tissue samples were spiked with a 0.5 µg/mL final concentration of test substances in two parallel measurements.

##### HPLC-FLD Analyses

The method was described in detail in a recent publication [24]. Briefly, the threefold volume of ACN was added to a 50 μL aliquot of plasma samples. After vortexing and centrifugation (10 min, 14,000× *g*, 4 °C), the supernatants were diluted with water then directly injected into the HPLC. Samples were analyzed using an HPLC system (Jasco, Tokyo, Japan), which included an autosampler (AS-4050), a binary pump (PU-4180), and a fluorescence detector (FP-920); the chromatograms were evaluated employing ChromNAV software. Samples (20 μL) were driven through a Security Guard™ (C_18_, 4.0 × 3.0 mm; Phenomenex, Torrance, CA, USA) guard column linked to a Gemini NX-C_18_ (150 × 4.6 mm, 3 μm; Phenomenex, Torrance, CA, USA) analytical column with a 1.0 mL/min flow rate, at room temperature. The eluent contained ACN and sodium phosphate buffer (10 mM, pH 7.0) (30:70 *v*/*v*% in eluent A, and 55:45 *v*/*v*% in eluent B). We applied 100% eluent A from 0 to 4 min, then the eluent was linearly changed to 100% eluent B from 4.0 to 4.5 min, after which 100% eluent B was used from 4.5 to 14.9 min. SZV 1287 and its metabolites (oxaprozin, L 2799, L 2805, and L 2811) were detected applying 320 and 368 nm excitation and emission wavelengths, respectively. Solutions applied for the calibration curves were freshly prepared and diluted every day. The calibration curves showed good linearity (R^2^ > 0.99).

#### 2.3.4. Thermophysiological Study

##### Experimental Design

First the effect of SZV 1287 on the proton-mode of TRPV1 receptor activation, the blockade of which is principally responsible for the hyperthermic effect of TRPV1 antagonists [31], was investigated. We detected Ca^2+^ influx in cultured primary sensory neurones of the trigeminal ganglia (TG) in response to proton-induced TRPV1 activation by microfluorimetry.

Following in vitro study, the potential side effect of SZV 1287 on deep body temperature was investigated using two different experimental setups in naïve male C57BL/6J mice. First, general locomotor activity (as a behavioral thermoeffector) and abdominal temperature (a form of deep body temperature) were measured with telemetry thermometry following a single administration of enterosolvent SZV 1287 (at 20 or 50 mg/kg, *n* = 7 and *n* = 8, respectively) or placebo capsules (*n* = 8).

In the next step, to exclude the masking effect of locomotor hyperactivity and the consequent abdominal temperature increase caused by the relatively stressful oral administration, deep body temperature (colonic) was measured with thermocouple thermometry following non-stressful i.p. infusion of SZV 1287 (at 20 or 50 mg/kg, *n* = 4 and *n* = 7, respectively) or its vehicle (*n* = 11).

##### Determination of Proton-Induced TRPV1 Receptor Activation in Cultured Primary Sensory Neurons

Ca^2+^ influx in cultured primary sensory neurons of the TG in response to proton-induced TRPV1 receptor activation was measured by microfluorimetry. Cultures were made from the TG of 1–3-day-old mouse pups. The ganglia were incubated for 35 min at 37 °C in phosphate-buffered saline (PBS) containing collagenase (type XI, 1 mg/mL) and then for 8 min with deoxyribonuclease I (1000 units/mL). After washing, the ganglia were dissociated by trituration, neurons were plated on poly-D-lysin-coated glass coverslips and grown in nutrient-supplemented medium containing 180 mL Dulbecco’s Modified Eagle Medium (D-MEM), where 20 mL horse serum, 20 mL bovine serum albumin, 2 mL insulin-transferrin-selenium-S, 3.2 mL putrescin dihydrochloride (100 μg/mL), 20 μL triiodo-thyronine (0.2 mg/mL), 1.24 mL progesterone (0.5 mg/mL), 100 μL penicillin, 100 μL streptomycin, and nerve growth factor (NGF, 200 ng/mL) was added. Cell cultures were maintained at 37 °C in a humid atmosphere with 5% CO_2_, as previously described [32].

Cell cultures of 1–3 days old were loaded for 30 min at 37 °C in a solution containing (in mM): NaCl, 122; KCl, 3.3; CaCl_2_, 1.3; MgSO_4_, 0.4; KH_2_PO_4_ 1.2; HEPES, 25; glucose, 10; pH 7.3, and 1 μM of fluorescent Ca^2+^ indicator dye, fura-2-AM (molecular probes). Dye-loaded neurons were washed with extracellular solution (ECS; containing in mM: NaCl, 160; KCl, 2.5; CaCl_2_, 1; MgCl_2_, 2; HEPES, 10; glucose, 10; pH 7.3) at room temperature. Calcium transients of TG neurons to ECS-pH 5.3 were examined with a fluorescence microscope (Olympus BX50WI, Olympus, Tokyo, Japan), as previously described [32]. ECS was gravity fed to the cells using a triple outlet tube of about 50 μm diameter from close vicinity. pH 5.3 and pH 5.3 + CZP arrived to the outlet via separate tubes (exposition time: 15 s). Rapid solution change and wash-out were controlled by a fast step perfusion system (Warner VC-77SP, Harvard Apparatus GmbH, March, Germany). Fluorescence images were taken with an Olympus LUMPLAN FI/x20 0.5 W water immersion objective and a digital camera (CCD SensiCam, PCO AG, Kelheim Germany) connected to a PC. Cells were illuminated alternately at 340 and 380 nm light generated by a monochromator (Polychrome II., TILL Photonics, Gräfelfing, Germany) under the control of Axon Imaging Workbench 2.1 software (AIW, Axon Instruments, Burlingame, CA, USA). Emitted light > 510 nm was measured. The R = F340/F380 was monitored (rate = 1 Hz) continuously, and the R values generated by the AIW software were then processed by the Origin software version 8.1 (Originlab Corp., Northampton, MA, USA). Baseline fluorescence was read from the period of recordings taken before exposing the cells to pH 5.3. Plates were pre-incubated with different concentrations of SZV 1287 (0.1, 1, and 10 μM) for 60 min, at 37 °C in a humid atmosphere with 5% CO_2_, or were untreated controls. The TRPV1 antagonist capsazepine (CZP; 10 μM) was co-administered with low pH.

##### Measurement of General Locomotor Activity and Abdominal Temperature

General locomotor activity and abdominal temperature were recorded with telemetry thermometry, as previously described [33]. Briefly, a miniature radiotransmitter (G2 E-Mitter series; Mini Mitter, Bend, OR, USA) was implanted in the mice under ketamine/xylazine (100/5 mg/kg, i.p.) anesthesia and gentamycin prophylaxis (22 mg/kg, i.m.). Through a small midline incision, the transmitter was inserted into the abdominal cavity and was fixed to the left side of the abdominal wall with suture. The surgical wound was closed, and mice were allowed to recover for at least 8 days before the measurement.

Following the recovery period, mice were studied inside their home cages, where they could freely move around. Telemetry receivers (model ER-4000; Mini Mitter) were positioned in a temperature-controlled (24 °C) room, and the cages were placed on top of the receivers. This setup allowed us to record changes of locomotor activity and deep body temperature after p.o. drug administration.

##### Measurement of Colonic Temperature

Colonic temperature was measured with thermocouple thermometry, as previously described [34]. Briefly, a small midline incision was made in the abdominal wall and a polyethylene-50 catheter filled with saline was implanted in the mice under ketamine/xylazine (100/5 mg/kg, i.p.) anesthesia and gentamycin prophylaxis (22 mg/kg, i.m.) 3–5 days before the measurement. The caudal end of the catheter was sutured to the right abdominal wall, the rostral end was heat-sealed, tunneled under the skin to the nape, and exteriorized. The catheter was flushed with saline on the day following the surgery and every other day thereafter.

Following the acclimation and recovery period, mice were placed in a cylindrical wire-mesh confiner allowing some back-and-forth movements but preventing the mice from turning around. Then, a copper-constantan thermocouple (Omega Engineering, Stamford, CT, USA) was inserted into the colon 3 cm proximal to the anal sphincter; it was fixed to the base of the tail with adhesive tape and plugged into a data logger (Cole-Parmer, Vernon Hills, OH, USA), which was connected to a computer. The exteriorized end of the i.p. catheter was connected to a polyethylene-50 extension filled with SZV 1287 or its vehicle consisting of 20% Kolliphor HS 15 (polyethylene glycol (15)-hydroxystearate, Sigma-Aldrich, St. Louis, MO, USA) and 80% distilled water. The extension was passed through a port of the climatic chamber (set to 30 °C) and connected to a syringe placed in an infusion pump (model 975; Harvard Apparatus, Mills, MA, USA), thus allowing the drug to be administered without disturbing the animal. The solutions (7.5 mg/mL) were prepared freshly before infusing via the i.p. catheter at a rate of 26 μL/min for 8 min.

### 2.4. Data and Statistical Analysis

Data are expressed as means ± standard errors of means (S.E.M.) and were analyzed by GraphPad Prism 8 software (GraphPad Software, San Diego, CA, USA). Decreases in mechanonociceptive withdrawal thresholds and changes in general locomotor activity and deep body temperatures were evaluated by repeated measures two-way ANOVA followed by Tukey’s multiple comparison test, while plasma concentrations of SZV 1287 and total plasma concentrations of SZV 1287 and its metabolites were evaluated by ordinary two-way ANOVA followed by Sidak’s multiple comparison test, and the percentage of proton-responsive cells and the change in the fluorescence ratio by one-way ANOVA followed by Dunnett’s multiple comparison test. *p* < 0.05 was considered significant.

## 3. Results

### 3.1. Nociceptive Study

#### 3.1.1. SZV 1287 in Enterosolvent Capsule Significantly Decreases Neuropathic Mechanical Hyperalgesia in Mice

The unilateral partial sciatic nerve ligation induced a significant, approximately 35–45% decrease in mechanonociceptive withdrawal thresholds compared to the baseline 7 days following the surgery in mice (Figure 1a). Single oral treatment of 20, 50, and 200 mg/kg SZV 1287 doses significantly, but not dose-dependently, reduced this neuropathic mechanical hyperalgesia 2 h later by approximately 50% (Figure 1a,b). SZV 1287 had no effect on the mechanonociceptive thresholds of the contralateral intact hind paws (Appendix A).

#### 3.1.2. SZV 1287 Is Absorbed from the Enterosolvent Capsule and Penetrates into the Brain

SZV 1287 together with its three main metabolites, such as oxaprozin, L 2805, and L 2811 was detectable in the plasma, although the parent compound was below the detection limit of the applied HPLC-UV method in cases of the 10, 20, and 50 mg/kg doses (Appendix A). Oxaprozin, L 2805, and L 2811 were already seen following the administration of the dose of 10 mg/kg (Figure 2a). The parent compound was only measurable following the administration of 200 mg/kg (0.59 ± 0.05 μg/mL), demonstrating the high presystemic elimination of SZV 1287 (Appendix A). In the brain, SZV 1287 was detectable above the dose of 20 mg/kg and its concentrations increased in parallel with the applied dose (Figure 2b). Except for very low concentrations of oxaprozin after the highest dose, other metabolites were not detectable in the brain (Appendix A).

#### 3.1.3. SZV 1287 Reaches Its Maximal Plasma Concentration 1 H after Oral Administration of Enterosolvent Capsules

SZV 1287 reaches its maximal concentrations in the plasma 1 h after oral administration of enterosolvent capsules in both fasted and non-fasted mice. Although the plasma concentrations of SZV 1287 were approximately doubled or tripled after the fasting compared to the non-fasted condition, the interindividual variations were much greater in the fasted group leading to statistically non-significant differences, when comparing the curves and the corresponding values at the different time points (Appendix A). The areas under the curves (AUC) values were 0.16 ± 0.09 in non-fasted and 0.38 ± 0.18 in fasted groups, which did not show a significant difference by the unpaired t test (*p* = 0.29). The total plasma concentration of SZV 1287 and its metabolites showed a maximum after 4 h in fasted mice and after 8 h in non-fasted animals; the total concentration was significantly higher in non-fasted mice at 8 h (Appendix A). However, the AUC values of the two curves (non-fasted: 20.71 ± 5.26, fasted: 13.7 ± 3.96) were not statistically significant (*p* = 0.31, unpaired t test).

### 3.2. Thermophysiological Study

#### 3.2.1. SZV 1287 Inhibits Proton-Induced TRPV1 Receptor Activation in Cultured Primary Sensory Neurons

Under control conditions, low pH (pH 5.3; high proton concentration)-evoked Ca^2+^-influx was detected in 31.39 ± 5.08% (30 neurons out of 97) of the cells, while the duration of the response until recovery was 49.9 ± 19 s, and the degree of recovery was 91.9 ± 12.8% (Figure 3a). A significantly decreased number of activated cells was detected 60 min after pre-incubation of the plates with 0.1, 1, and 10 µM SZV 1287, as well as after co-administration of the TRPV1 antagonist CZP (10 µM). The respective values after pH 5.3 administration were 17.14 ± 5.8% (14 out of 77), 17.87 ± 5.48% (14 out of 88), 9.86 ± 4.08% (7 out of 65), and 7.06 ± 5.06% (6 out of 85) (Figure 3a). The respective durations until recovery were 43.7 ± 16.8 s, 39.9 ± 15.8 s, 33.16 ± 8.85 s, and 31.33 ± 10.78 s, and the respective degrees of recovery were 93 ± 15.8%, 92 ± 10.8%, 97.15 ± 9.8%, and 96.6 ± 5.6%. Similarly to CZP, 10 µM SZV 1287 pre-treatment significantly decreased the number of proton-activated cells. The R value demonstrating the change of the fluorescence ratio of individual neurons was also significantly decreased by both 1 and 10 µM SZV 1287, as well as by 10 µM CZP (Figure 3b).

#### 3.2.2. SZV 1287 Does Not Substantially Affect General Locomotor Activity and Deep Body Temperature Following Enterosolvent Capsule Administration in Mice

Similarly to placebo, the enterosolvent SZV 1287 capsules exerted no effect on the increased general locomotor activity and abdominal temperature induced by the relatively stressful oral administration procedure in mice. After this transient increase, both parameters decreased and remained statistically indistinguishable until the end of the experiment (Figure 4a,b).

Since in the telemetry thermometry setup the substance administration procedure may lead to stress-induced hyperkinesis and consequent hyperthermia (as also observed in the placebo-treated group; see Figure 4a,b) masking small changes in thermoregulatory responses, we also studied the effects of SZV 1287 in the thermocouple setup following non-stressful infusion through a pre-implanted i.p. cannula. Infusion of SZV 1287 at 20 mg/kg had no remarkable effect on deep body temperature compared to the placebo (Figure 4c). When SZV 1287 was infused at 50 mg/kg i.p., it caused a tendentious, but not significant, increase in the colonic temperature, which started to increase 40 min after the infusion and remained elevated until 120 min (*p* = 0.93, 0.36, 0.25, 0.28, 0.35, 0.29, 0.23, 0.17, and 0.17 at 40, 50, 60, 70, 80, 90, 100, 110, and 120 min, respectively, Figure 4c).

## 4. Discussion

We provide here the proof-of-concept that our novel multi-target drug candidate SZV 1287 administered in an enterosolvent preparation planned for the phase I clinical trial inhibits neuropathic mechanical hyperalgesia in mice. It rapidly penetrates the brain as shown by HPLC analysis supporting our previous PET/MR imaging data [22], which suggests at least partially a central mechanism of action involved in its analgesic effect. We demonstrated its thermoregulatory safety despite its ability to antagonize the TRPV1 receptor [10].

SZV 1287 was patented in 2010 together with other AOC3 inhibitors for the prevention and treatment of diseases related to acute or chronic inflammation, carbohydrate metabolism including diabetes-associated complications (retinopathy and macular edema), adipocyte or vascular endothelial and smooth muscle functions, as well as neurodegeneration [9]. The unique feature distinguishing this compound from other AOC3 inhibitors is that it was designed on the basis of a novel, innovative drug design strategy termed as metabolism-activated multitargeting (MAMUT) [8]. SZV 1287 itself irreversibly inhibits AOC3, and its active metabolite oxaprozin inhibits COX. The combination of these two synergistic anti-inflammatory mechanisms raised the opportunity for the development of a conceptually novel anti-inflammatory drug [8]. Its greater anti-inflammatory effect was also demonstrated in both carrageenan-induced acute and complete Freund’s adjuvant (CFA)-induced chronic inflammation compared to the reference selective AOC3 inhibitor LJP 1207 [21].

Since the highly reactive end-products of AOC3 such as formaldehyde, methylglyoxal, and hydrogen peroxide are known TRPA1 activators [15,16], our group first addressed the potential analgesic effect of AOC3 inhibitors explained by diminished TRPA1 receptor activation on the nociceptive neurons. The antihyperalgesic effect was evidenced in the partial sciatic nerve ligation model of traumatic neuropathy after i.p. administration [20] leading to our next patent focusing on the analgesic effect of SZV 1287 under neuropathic conditions [5]. Based on the similarity of the chemical structure of SZV 1287 with some oxime derivative TRPA1 antagonists [35,36,37], we investigated its effect on allyl-isothiocyanate-induced TRPA1 and capsaicin-induced TRPV1 activation in cultured TG neurons, as well as on TRPA1 or TRPV1 receptor-expressing cell lines. The results revealed that SZV 1287, but not LJP 1207, significantly and concentration-dependently decreased both TRPA1 and TRPV1 activation, demonstrating its direct dual antagonistic property on these receptors [10]. These mechanisms are likely to contribute to the analgesic effect of SZV 1287, since the role of these receptors is well-known in nociception, particularly in neuropathic pain [38,39]. Because of this particularly interesting and favorable multi-target mode of action, SZV 1287 became the lead molecule of our further investigations, in which we proved its antihyperalgesic effect in two different chronic arthritis models and clarified the role of TRPA1 and TRPV1 receptors in its mode of action after i.p. administration [19,20]. Furthermore, we demonstrated its beneficial effects in a chronic insulin-controlled diabetes model including protection against streptozotocin-induced beta cell damage and some microvascular complications, such as neuropathy and retinopathy [23].

We initiated the drug development process of SZV 1287 with the main indication of neuropathic pain in 2016 and have recently submitted the preclinical documentation containing chemical, analytical, pahrmaceutical technological, pharmacodynamic, pharmacokinetic, and safety/toxicological profiling of the compound and the enterosolvent formulation aimed to be used in the phase I clinical trial. We demonstrated here that SZV 1287 significantly, but not dose-dependently, reduces neuropathic hyperalgesia by 50% of its maximal effect even after a single treatment in enterosolvent capsules. This result is in agreement with our previous findings after i.p. administration in the same model [20]. The lack of dose-dependency is not surprising in this experimental setup [20,27], since the nocifensive behavioral outcome parameter (mechanonociceptive withdrawal threshold) is the result of complex peripheral and central sensitization mechanisms. Furthermore, it may also be explained by the multi-target and irreversible AOC3 inhibitory mechanism of action of SZV 1287. It cannot be excluded that in higher doses SZV 1287 acts at other target(s), which stimulate the ascending nociceptive pathway or inhibit the descending pain inhibitory pathway leading to a plateau effect. The irreversible AOC3 inhibitory mode of action resulting in complete enzyme inhibition already in the minimal therapeutic dose can also explain this plateau effect.

The success of the enterosolvent formulation is justified by both our consistent antihyperalgesic results and the stable absorption of SZV 1287. Because of its instability under acidic conditions, it would likely be degraded in the stomach without the enterosolvent coating leading to unpredictable absorption and pharmacodynamic consequences. The results of our exploratory pharmacokinetic study suggest that SZV 1287 may produce its antihyperalgesic effects rapidly within a few hours following administration of the enterosolvent capsule. Furthermore, its high presystemic elimination was also demonstrated, because only low concentrations of SZV 1287 reached the systemic circulation compared to the metabolites, mainly oxaprozin. However, the observed inhibitory effect is unlikely to result from oxaprozin, but SZV 1287 itself, because oxaprozin was previously ineffective in the same model [20]. We also showed the brain penetration of SZV 1287 with HPLC analysis supporting our previous PET/MR imaging data [22]. Although brain concentrations were relatively low in all effective SZV 1287 doses, these were sufficient to produce an analgesic effect, which could be explained by the irreversible mode of action of SZV 1287 resulting in complete AOC3 inhibition already in the detected low concentrations. The limitation of this preclinical study is that because of the need for specific enterosolvent capsules formulated the same way as for the clinical trials and the unavoidable variability range of the mouse body weights, exact dosing of SZV 1287 in all animals belonging to an experimental group was not technically possible. We had to accept minimal alterations of the applied doses within a group.

Since SZV 1287 enters the brain and has TRPV1 antagonistic action, we investigated its thermoregulatory safety. It is well-known that TRPV1 receptors play a key role in thermoregulation [40,41], and the development of several TRPV1 antagonists failed both in preclinical and clinical phases because of their hyperthermic effect [42]. Therefore, the drug authorities require the preclinical investigation of hyperthermia regarding novel TRPV1 antagonists. Since the blockade of proton-mode of TRPV1 activation is principally responsible for this effect [31], first we investigated the effect of SZV 1287 on proton-induced TRPV1 activation. SZV 1287 significantly decreases the proton-induced TRPV1-mediated calcium-influx in cultured primary sensory neurons similarly to the capsaicin-induced responses [10] raising the potential risk for hyperthermia. However, SZV 1287 did not substantially influence the deep body temperature in two different experimental paradigms. In the telemetry thermometry setup, both placebo and SZV 1287 increased locomotor activity leading to deep body temperature increase after oral treatment, which is likely to be the result of the stressful oral administration procedure. In the thermocouple thermometry setup, in which such stress-induced alterations are much less, i.p. infused SZV 1287 did not cause a significant colonic temperature increase at the minimal analgesic dose of 20 mg/kg but induced a slight increase at the highest applied dose of 50 mg/kg. However, the thermoregulatory effects of TRPV1 antagonists in animal models should be cautiously interpreted to humans, as inter-species differences in the activation mechanisms of TRPV1 receptor have recently been described [41].

## 5. Conclusions

In conclusion, oral SZV 1287 inhibits neuropathic mechanical hyperalgesia in enterosolvent formulation and, despite being a centrally acting TRPV1 receptor antagonist, it does not substantially influence thermoregulation in mice, which makes it a promising analgesic candidate.

## Figures and Tables

**Figure 1 biomedicines-09-00749-f001:**
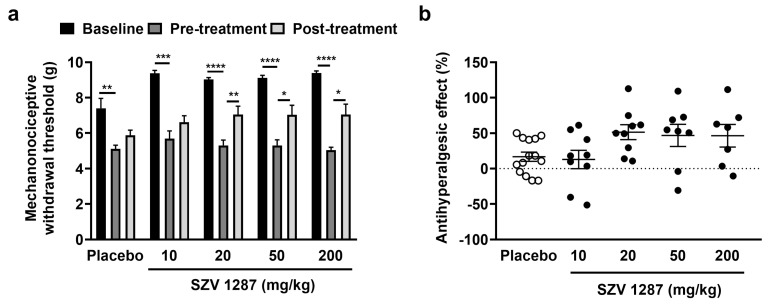
Effect of SZV 1287 on partial sciatic nerve ligation-induced mechanical hyperalgesia following oral dosing in mice. (**a**) Mechanonociceptive withdrawal thresholds and (**b**) antihyperalgesic effects calculated as ((percentage changes of pre-treatment mechanonociceptive withdrawal thresholds compared to the initial controls—percentage changes of post-treatment mechanonociceptive withdrawal thresholds compared to the initial controls)/percentage changes of pre-treatment mechanonociceptive withdrawal thresholds compared to the initial controls) × 100 2 h following single enterosolvent SZV 1287 capsule treatment (10, 20, 50, and 200 mg/kg) on the 7th postoperative day. Data are shown as means ± S.E.M. of *n* = 7–14 mice/group, * *p* < 0.05, ** *p* < 0.01, *** *p* < 0.001, **** *p* < 0.0001; repeated measures two-way ANOVA (baseline, pre-treatment, and post-treatment factor F = 137.7, *p* < 0.0001) followed by Tukey’s multiple comparison test.

**Figure 2 biomedicines-09-00749-f002:**
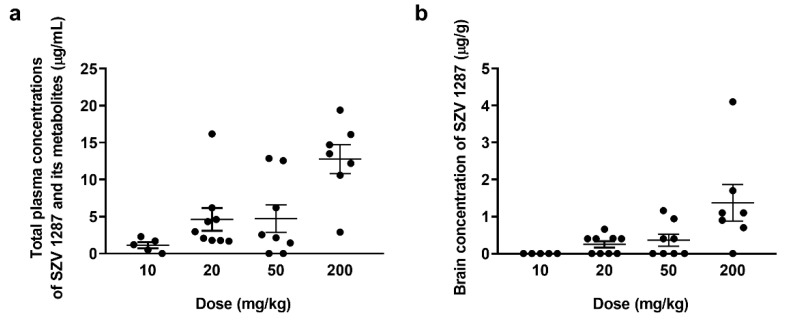
(**a**) Plasma concentrations of SZV 1287 and its main metabolites (oxaprozin, L 2805, L 2799, and L 2811) and (**b**) brain concentrations of SZV 1287 following single enterosolvent capsule treatment (10, 20, 50, and 200 mg/kg) in mice. Data are shown as means ± S.E.M. of *n* = 5–9 mice/group.

**Figure 3 biomedicines-09-00749-f003:**
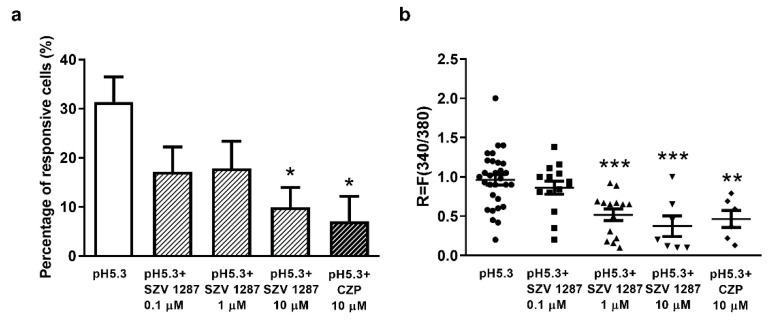
Effect of SZV 1287 on proton-induced TRPV1 receptor activation on cultured primary sensory neurons. (**a**) The percentage of proton-responsive cells at pH 5.3 or pH 5.3 + 10 µM CZP is presented on control plates and on SZV 1287-pretreated (0.1, 1, and 10 µM) plates. Ca^2+^-responses are presented in % of total number of examined neurons, * *p* < 0.05 vs. control; one-way ANOVA (treatment factor F = 3.45, *p* = 0.017) followed by Dunnett’s multiple comparison test. (**b**) The change in the fluorescence ratio (R = F340/F380) demonstrating the activation of individual cells is presented on the control, SZV 1287-pretreated (0.1, 1 and 10 µM) and CZP-treated (10 µM) plates. Each column represents mean + SEM of *n* = 65–97 cells/group, ** *p* < 0.01, *** *p* < 0.001 vs. the control; one-way ANOVA (treatment factor F = 8.73, *p* < 0.0001) followed by Dunnett’s multiple comparison test. Abbreviation: CZP, capsazepine.

**Figure 4 biomedicines-09-00749-f004:**
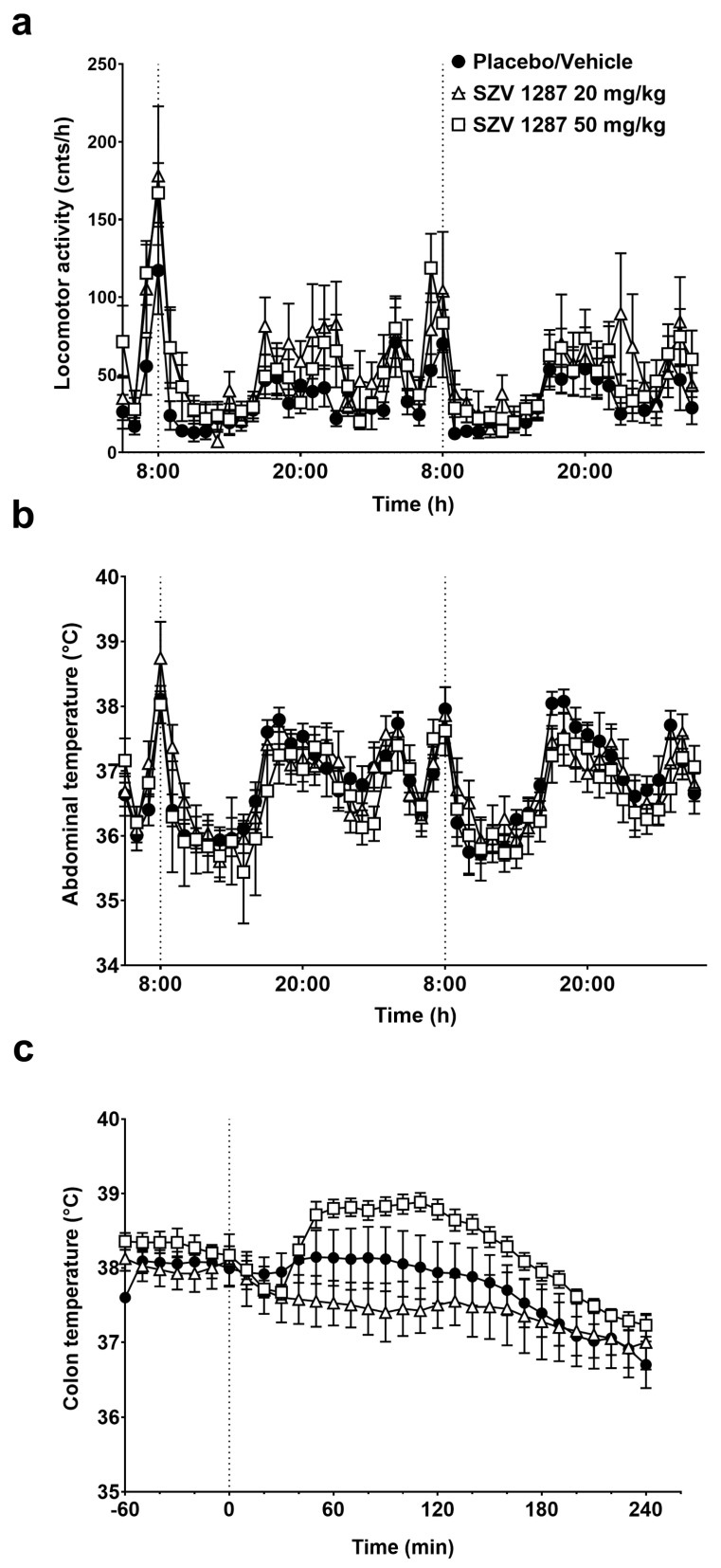
Effects of SZV 1287 on general locomotor activity and deep body temperature in mice. (**a**) General locomotor activity and (**b**) abdominal temperature in response to a single administration of enterosolvent capsules; (**c**) colonic temperature in response to single i.p. administration of 20 and 50 mg/kg doses of SZV 1287 compared to the placebo- or vehicle-treated group. Data are shown as means ± S.E.M. of *n* = 4 in the 20 mg/kg i.p., *n* = 7 in the 20 mg/kg p.o. and 50 mg/kg i.p., *n* = 8 in the placebo and 50 mg/kg p.o., *n* = 11 mice in the vehicle-treated groups (repeated measures two-way ANOVA followed by Tukey’s multiple comparison test).

## Data Availability

The data presented in this study are available on request from the corresponding author. The data are not publicly available since the complete preclinical dossier together with phase I clinical study protocols have just been submitted to the national drug authority. The raw data are confidential.

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
