# Peer review of "Proof-of-Concept for the Analgesic Effect and Thermoregulatory Safety of Orally Administered Multi-Target Compound SZV 1287 in Mice: A Novel Drug Candidate for Neuropathic Pain"

_biomedicines, 2021, doi:10.3390/biomedicines9070749_

Round 1
Reviewer 1 Report
The manuscript “Proof-of Concept for the Analgesic Effect and Thermoregulatory Safety of Orally Administered Multi-Target Compound SZV 1287 in Mice: A Novel Drug Candidate for Neuropathic Pain” is an another preclinical study that investigates this multitarget compound as an antinociceptive drug for treatment of neuropathic pain. This topic is important since currently new non opioid analgesics are needed. Authors used different in vivo and in vitro approaches for studies effectiveness and safety of SZV 1287 and demonstrated that this substance could be a useful for treatment of neuropathic pain. Authors well introduced the scientific topic of the research, used appropriate statistical methods for analysis, and appropriately discussed obtained results. However, some improvement is needed to increase quality of the paper.
- Page 2, line 52: "Neuropathic pain of 317% prevalence represents a great unmet medical need…" the sentence must be edited.
- Page 3, line 99: "The ability of SZV 1287 to inhibit neuropathic mechanical hyperplasia suggests that it exerts central mechanism of action..." The reduction of neuropathic hyperplasia following i.p or intraoral drug administration is not a strict evidence of central analgesia since peripheral effect on sensitized afferents during neuropathic pain is also possible and well described.
- Page 3, line 134: “…demonstrated by the 1H NMR study…” the abbreviation should be explained.
- Page 6, line 277: Experimental Model. 1) As I understand all in vivo experiments were performed in mice after the partial sciatic nerve ligation. For clarity this should be mentioned during explanation of obtained results of experiments. 2) Sham operated group is needed for this research firstly to clearly demonstrate effectiveness of ligation and secondly to understand does SZV 1287 affects only prolong pain or also physiological nociceptive sensation.
- Page 6, line 291: Measurement of Touch Sensitivity Thresholds of the Hind Paws. This method is better to describe as a measurement of nociceptive mechanical withdrawal threshold.
- Page 7, line 337: “Baseline fluorescence was read from the period of recordings taken before exposing the cells to pH 5.3.” It is not clear how solution with pH 5.3 was exposed to cells, for how long, and was the effect washed out after applications?
- Page 8, line 366, Measurement of Colonic Temperature. It will be better to describe implantation of i.p. cannula before Experimental Design section. In this case first appearance of "i.p. cannula" (page 8 line 352) will be understandable. It is also not clear do naïve mice or mice with neuropathic pain were used for measurement of deep temperature. Of course, it will be better to use groups of sham operated and neuropathic mice.
- Figure 1a and b: The demonstration of effects in g and % is a repetition of data which complicates reading. I think demonstration of raw mechanical threshold (g) is preferable.
- Figure 2b: If so low brain concentration of SZV 1287 occurred and only after administration of high doze, how central effect could be developed?
- Figure 4c: It is looks strange that the colon temperature following the placebo and SZV 50mg/kg is not differ. This could be a result of big differences in experimental groups. It is suggested to indicate the number of animals for each experimental group and show results of two-way ANOVA in the text. However, the best way is to make groups sizes comparable. I also suggest making size of symbols bigger in Figure 4, because it is hard to distinguish between SZV 20 and SZV 50.
Reviewer 2 Report
The authors describe the preliminary preclinical characterization of the multi-target compound SZV 1287 in a mice model by mean of pharmacological and pharmacokinetic assays. Despite the paper would have been potentially interesting to the readers, there are some major flaws that need to be corrected before the paper is acceptable for publication.
1) Methods should be adequately described. For instance it is unclear to the reader which is the amount of SZV administered to any single mice. The authors state that "studies were performed on 8–12 week-old 156
male NMRI (weighing 35–45 g) and C57BL/6J mice (weighing 20–30 g)". Why using two different animal strain?
2) As for the point 1: considering the differences in weight and considering that SVZ was administered via capsules containing a fixed amount of drug, how could the authors state that capsules filled with 0.4 mg SZV 1287 correspond to the 10 mg/kg dose while capsules filled with 0.8 mg SZV correspond to the 20 mg/kg? The capsules roughly correspond to the reported doses, but the experimental setup infringes the rule of equivalent dose, that is mandatory for a animal study. Moreover the authors administered. Moreover the authors state: “In case of the 50 and 200 mg/kg doses two and four capsules containing 1 mg SZV 1287 were administered". The effect of a single administration could not be compared with a multiple administration regimen. The resulting data cannot be compared in any case, being subjected to extended variability.
3) Why two different analytical methods (HPLC-UV and HPL-FLD) have been useed for the assessment of drug plasma concentration? Which are the differences? Which method has been used to populate the data table?
4) Pharmacokinetic experiments should be more clearly described. Which are the equations deriving from the analytes calibration curves? Which is the amount of blood taken? At which timepoint the blood was taken after capsule administration?
5) It is unclear to me how the blood peak plasma concentration was calculated. Please explain
6) When pharmacological measurements were performed? Please describe the lag time between capsule administration and measurements
7) It is hard to figure out that a single oral administration is responsible for such clear pharmacological effect, especially at the presence of the reported blood and brain concentration. This point must be fully explained. The same authors claim a central effect for the molecule, but brain concentrations are really low? Do the authors believe the right analytical method has been used? Probably HPLC/MS/MS methods would have been more suitable.
8) Why not using oral gavage, instead of capsule administration?
9) Does SZV revert only proton-induced TRPV-1 activation? Did the authors challenged the drug effect at the presence of a canonical TRPV1 agonist?
Reviewer 3 Report
The Authors highlight the role of a new candidate molecule with a dual mechanism of action TRPV1 and TRPA1 antagonist, in the treatment of neuropathic pain. The molecule was already tested in the inflammatory pain models by the Authors.
The paper is interesting and the results support the conclusion of the Authors.
Some issue are related to statistic and general aspects:
The Authors should add the F-values for further strengthen the statistical significance of the results.
Do the oral administration stressed the mice? is this related to the analgesic effect?
English should be revised
Round 2
Reviewer 1 Report
The manuscript titled: “Thermoregulatory Safety of Orally Administered Multi-Target Compound SZV 1287 in Mice: A Novel Drug Candidate for Neuropathic Pain” is significantly improved after the revision. However, one concern is remained, and it affects readiness of the paper for publication.
In the subsection of the Method: “Determination of Proton-Induced TRPV1 Receptor Activation in Cultured Primary Sensory Neurons” authors noted that “Rapid solution change and wash-out were controlled by a fast step perfusion system…” In the text of the results and in the Figure 3 effects of pH 5.3, SZV, and CZP are well presented, however data about recovery of the effects is missing. Absence of data about effects recovery following drugs wash-out limits value of obtained results. Timing and degree of recovery are valuable characteristics of functional effects, and they are especially important for the study of a potential drug candidates. I am asking to add these data before publication of the paper.
Reviewer 2 Report
Kind authors,
here follows a point-by-point comment to your reply, to further clear my point of view:
1) The methods have been more precisely described as detailed below.
The thermophysiological study was designed after completing nociceptive experiments in NMRI mice, which is a widely used strain for these experiments. The reason why we chose C57BL/6J strain for thermophysiological study is if SZV 1287 had induced significant hyperthermia in this strain we would have investigated it in TRPV1-deficient mice as well generated on this background to explore the involvement of this receptor.
Comment: The potential use of a knock-out strain does not justify the use of two different strains. Nociceptive experiments could have been easily performed over C57BL/6J, as for literature evidences. The use of two different strains introduces variability leading to a difficult interpretation of the results.
2) We stated that the substance amounts were calculated for a mouse weighing 40 g. They were corrected according to the average weight of mice enrolled into both nociceptive and thermophysiological experiments. It was clarified in the revised manuscript (page 3, line 126-130). Therefore, administering the same capsule resulted in a maximum of ±2 mg/kg dose difference in some cases, which we consider to be an acceptable variability and representative of the real life use of human oral preparations. This is supported by our previous results obtained in the same model after i.p. administration of SZV 1287 showing similar 50% antihyperalgesic effects in response to 5, 10 and 20 mg/kg doses (Horváth et al., Pharmacol Res, 2018). Since the size of capsules was limited, we had to apply multiple administration regimen to investigate the effect of higher doses. We did not observe any behavioral changes or other variabilities between the groups treated with either one or more capsules. However, we agree with the reviewer that the use of “single treatment” would be more adequate instead of “single administration”. We corrected this term throughout the revised manuscript.
Comment: ±2 mg/kg is not exactly considered an acceptable bias during preclinical evaluation. It also means a difference of about 4mg/Kg from the lowest to the highest weight. And this usually makes a difference. I agree with the authors that these differences are experienced in the typical posology of drugs but these drugs have been clinically challenged. In preclinical assessment the same amount of drug should be mndatorily given to the animals, othervwise, results are confusing. The use of multiple administrations is another variable introduced. Dissolution rate of a single capsule is different from the dissolution rate of two capsules. Comparison are almost impossible considering the different bioavailabilities.
3) The different methods were applied due to technical issues, but it did not affect our main results and conclusions as shown by the results in Figs. 2.a. and S4.b. The two techniques provided roughly the same plasma concentrations of SZV-1287 and its main metabolites. Most of the tissue samples were analyzed by HPLC-UV method. Nevertheless, the plasma levels of SZV 1287 and some of its metabolites are very low in the circulation. Therefore, we felt reasonable to also quantify the plasma concentrations employing a more sensitive analytical method than HPLC-UV. Since SZV 1287 and its metabolites exert strong fluorescence, we were able to quantify these compounds in the circulation with much higher sensitivity (compared to HPLC-UV), employing HPLC-FLD technique. During the optimization of the HPLC-FLD method, we used a different HPLC system, and we had to consider not only the retention times but the fluorescence signal of the fluorophores in the eluent applied. Therefore, we employed a different HPLC assay for the fluorescent, highly sensitive quantification of SZV 1287 and its metabolites.As we showed in the corresponding Results section (page 10, line 441) and in the table legends, these data were derived from HPLC-UV analysis.
Comment: This is even more confusing. You decided to use the UV detector realizing it was unfit to detect the blood concentration of SZV1287. Thus, you decided to use a more sensible detector and, supposedly, you performed a brand new standardization. Nevertheless results are derived from HPLC-UV analysis.
4) Plasma and brain concentrations were quantified by HPLC methods, employing calibrations curves after the linear fitting to the concentration-AUC curves (see the representative data below). Solutions applied for the calibration curves were freshly prepared and diluted every day, which showed good linearity (R2 > 0.99). This information was inserted into the revised manuscript (page 6, line 258-259 and page 7, line 312-313). 150-200 L blood was taken to analyze plasma concentrations of SZV 1287. We inserted this information into the revised manuscript (page 4, line 185 and page 5, line 204). As we described in the corresponding Methods section, in the exploratory pharmacokinetic study blood was taken 1, 2, 4, 8 and 24 h, in the nociceptive study 2 h following the treatment (page 4, line 184-185 and page 5, line 203).
Comment: The question is: since authors cannot correlate the pharmacokinetic parameters to the biological effects, which is the aim of the pharmacokinetic study? Why not using more sensible analytical methods in order to establish a clear and concise correlation between drug bioavailability and biological effects? How could the data presented be used by other researchers? Which is the need for describing drug metabolites determination in this context?
7) We described in the corresponding Methods section that following the posttreatment nociceptive measurements blood was taken by cardiac puncture and brains were removed to analyze the plasma and brain concentrations of SZV 1287, respectively (page 5, line 203-206). SZV 1287 was detectable in the brain above the minimal analgesic dose of 20 mg/kg and its concentrations showed dose-dependent increase. The relatively low brain concentrations of effective SZV 1287 doses were sufficient to produce analgesic effect, which could be explained by its irreversible inhibition of the AOC3 enzyme. This is likely to result in a consequently reduced production of tissue irritant mediators activating -among others - TRPV1 and TRPA1 receptors. This explanation was inserted into the Discussion section (page 14, line 586-589). We agree that the HPLC/MS/MS method is more sensitive, but these techniques were available and routinely applicable after optimization for the measurement of SZV 1287 and its metabolites in the present studies, and the results provided clear outcomes.
Comment: please see the comment to point 4. I strongly believe that the results given do not provide clear outcomes. On the other hand the experimental procedures deeply penalise a potentially interesting paper.
8) We used the enterosolvent capsule formulation instead of oral gavage to prevent the unfavorable biotransformation to oxaprozin under acidic condition in the stomach (oxaprozin was not active against neuropathic pain as described in our previous publication, Horváth et al., Pharmacol Res, 2018). The same enterosolvent (acid resistant) preparation is planned to be tested in the clinical trials and we aimed to provide translational results during the preclinical investigations in rodents. We described this explanation in both Introduction (page 3, line 108-109) and Discussion sections (page 13, line 519-520).
Comment: Oral gavage does provide protection from the stomach acid environment and provides the administration of equimolar doses of the drug to each animal group.
